# A Fumigation-Based Surface Sterilization Approach for Plant Tissue Culture

**DOI:** 10.3390/ijerph18052282

**Published:** 2021-02-25

**Authors:** Iyyakkannu Sivanesan, Manikandan Muthu, Judy Gopal, Shadma Tasneem, Doo-Hwan Kim, Jae-Wook Oh

**Affiliations:** 1Department of Bioresources and Food Science, Konkuk University, 1, Hwayang-dong, Gwangjin-gu, Seoul 05029, Korea; isivanesan@gmail.com (I.S.); kimdh@konkuk.ac.kr (D.-H.K.); 2Laboratory of Neo Natural Farming, Chunnampet, Tamil Nadu 603 401, India; bhagatmani@gmail.com (M.M.); jejudy777@gmail.com (J.G.); 3Department of Chemistry, Faculty of Science, Jazan University, Jazan P.O. Box 114, Jazan 45142, Saudi Arabia; sthaque@jazanu.edu.sa; 4Department of Stem Cell and Regenerative Biology, Konkuk University, Seoul 05029, Korea

**Keywords:** fumigation, antimicrobials, plant tissue culture, surface sterilization, carbon nanodots

## Abstract

Plant tissue culture has led to breakthroughs in understanding and applying the fundamental knowledge towards harnessing more from plants. Microbial contamination is one of the serious problems limiting the successful extrapolation of plant tissue culture practices. Sources of in vitro contamination include culture containers, media, explants, equipment, the environment of the culture room and transfer area, and operating personnel. The successful initiation of in vitro culture mostly depends on surface sterilization of explants because this is the primary source. Usually, surface sterilization is done using chemicals, or toxic nanomaterials, this is the first time such an approach has been demonstrated. Numerous surface microflora attached to plant surfaces grow faster than the cultured explants and release phytotoxic substances into the culture media, hindering positive outcomes. In the current work, for the first time, the applicability of turmeric and benzoin resin-based fumigation of explants is demonstrated. The results showed that fumigation methods for surface sterilization were promising and could lead to fifty and even 100% contamination-free plant tissue culture. Nanoparticulate carbon was identified in the turmeric and benzoin smoke and coined the key player in the surface sterilization effect. These studies open a whole new avenue for the use of fumigation-based methods for riddance of microbial contamination.

## 1. Introduction

Plant tissue culture involves cultivation of explants on artificial media in a controlled laboratory environment. A supportive culture medium and environment for the explants is crucial for the success of this propagation technique. Further, culture media manipulations and cultural techniques eventually enable the regeneration of challenging plant species in vitro from cultured explants via plant biotechnological tools [1,2,3]. In vitro plant culture methods require expensive production facilities and skilled persons and face several problems associated with in vitro culture establishment, such as explant browning, hyperhydricity, microbial contamination, shoot tip necrosis, and tissue proliferation.

Microbial contamination is one other challenge facing plant tissue culture practices. Several possible sources of in vitro tissue culture contamination are culture containers, media, explants, equipment, the environment of the culture room and the transfer area, and operators [4,5,6,7,8,9]. The successful initiation of in vitro culture rests on effective surface sterilization of explants. This is crucial, since a wide range of microbes are attached to the surface of plants, and these grow faster than the cultured explants and release phytotoxic substances into the culture media, interfering with the culturing process. Several chemicals have been used to disinfect the explants before culturing [4]. However, surface sterilants are also toxic to the explants, and the level of toxicity depends on the dose of disinfectants and duration of treatment. Some sterilants have even failed to eliminate contaminants in explants; rather, they profoundly affect morphogenesis due to their phytotoxicity. Removing endophytic microbes is very challenging, and hence, antibiotics and antifungal compounds have been included in plant tissue culture media to destroy or prevent microbial growth [10]. Several studies have reported the phytotoxicity of antimicrobial agents to cell, organ, and tissue cultures in plant species [5,6,7,8,10].

In the recent decade, nanoparticles (NPs) have been applied for eliminating plant tissue culture contaminants [9]. The explants of *Araucaria excelsa*, *Cynodon dactylon*, *Gerbera jamesonii*, olive, *Valeriana officinalis,* and *Vitis vinifera* obtained from greenhouse- or field-grown plants were disinfected with ethanol or commercial bleach, and then silver nanoparticles (AgNPs) eventually reduced or eliminated the contaminants [11,12,13,14,15]. Few reports have confirmed that Ag NP treatments led to successful surface sterilization of cotyledons, leaves or seeds of *Arabidopsis*, G × N15 (hybrid of almond × peach) rootstock, potato, *Rosa hybrida,* and tomato [16,17,18,19]. Inclusion of Ag NPs, TiO_2_ NPs, Zn NPs or ZnO NPs in the culture medium is reported to reduce internal and external contamination in *Araucaria excelsa*, *Bacopa monnieri*, banana, barley, G × N15 (hybrid of almond × peach) rootstocks, olive, potato, *Rosa hybrida*, *Vanilla planifolia,* and tobacco [12,16,18,19,20,21,22,23,24,25,26,27]. The effectiveness of NPs in the elimination of microbial contaminants in plant tissue culture depends on their dimension, size, distribution, and type. Several researchers have also reported adverse effects of NPs on the survival and subsequent regeneration of explants while using cytotoxic NPs [9].

For the first time, we report the use of fumigation for surface sterilization of plant explants. Surface sterilization of plant explants has usually been accomplished using disinfectant solutions or nanomaterials. The impact of these sterilants is considered an unavoidable necessity. However, for the first time, we have demonstrated a technique where biocompatible fumigants from plant sources have been used for surface sterilization of explants. Turmeric and benzoin resin were used as the fumigation sources. The methodologies for fumigation that have been used in this work are also unique and reported for the first time. The role of carbon nanoparticles in the surface sterilization effect and the characterization of the carbon nanoparticles in the fumigants are also reported. This study opens up a new avenue of research towards the discovery of various other possible biocompatible fumigation methods offering ‘green’ (plant-based) solutions for surface sterilization of plant explants.

## 2. Materials and Methods

### 2.1. Preparation of Explants

Actively growing shoots of three ornamental plants (in the flowering stage), *Rhododendron yedoense* var. *Poukhanense*, *Hedera helix*, and *Rosa hybrida* ‘Red Sandra’, were collected from Konkuk University campus, Gwangjin-gu, Seoul, Korea in the month of June. They were thoroughly washed under running water. The shoot tips were defoliated, washed with mild detergent solution, rinsed with sterile distilled water for 1 min, and then blotted dry using sterile filter paper to remove traces of water. *Rhododendron yedoense* var. *Poukhanense* was coded as Explant 1, *Hedera helix* as Explant 2, and *Rosa hybrid* as Explant 3 in the subsequent sections.

### 2.2. Fumigation-Based Surface Sterilization

Turmeric and benzoin resin (Indian local name: sambrani), procured from the local supermarket in Tamilnadu, India, were used for the fumigation experiments. The explants were subjected to fumigation following three methods. Method A: fumigation of tissue culture media plates for a period of 3 min, by holding the agar plate (at a distance of 15 cm) in an inverted position over the fuming turmeric rhizome (Tur) or benzoin resin (BR). The culture media plates contained Murashige and Skoog medium amended with 0.8% (*w/v*) plant agar and 3% (*w/v*) sucrose. Method A thus involved fumigation of the culture medium in the plates prior to seeding the explants. The fumigated plates were kept sealed until use. Method B: Direct fumigation of explants held (at a distance of 30 cm) over fuming Tur/BR for 10 sec. Method C: The Tur/BR smoke as collected in sterile glass bottles and the settled soot dispersed in 25 mL of 20% ethanol (EtOH) (20 mL of ethanol in 80 mL of sterile distilled water); the explants were pretreated by immersing them in this soot solution for 5 min. Single nodal segments (one inch long) were directly placed on the plates prepared according to Method A. Single nodal segments (one inch long) prepared from Method B and C were inoculated on Petri dishes containing Murashige and Skoog (MS) medium. Control sets, which involved non-fumigated media plates and un-pretreated explants, were prepared for comparison. The experiment was conducted in triplicate with 25 explants for each treatment. Assessment of bacterial contamination was conducted two days after incubation and for fungal contamination, after a week’s incubation. The culture medium consisted of MS nutrients and vitamins [28] amended with 0.8% (*w/v*) plant agar and 3% (*w/v*) sucrose. The pH of the medium was adjusted to 5.6 before adding agar and was sterilized at 121 °C for 20 min. The cultures were kept at 25 ± 2 °C under a 16 h photoperiod with a photosynthetic photon flux density of 45 µmol m^−2^ s^−1^. Figure 1 presents the experimental details.

### 2.3. Characterisation of the Fumigants

The Tur/BR smudges suspended in 20% ethanol were characterized for the presence of carbon nanomaterial using a spectrophotometer (Nanodrop ND-1000 v 3.3.1), (Nanodrop Technologies, Inc., Wilmington, NC, USA). The absorbance was scanned from 220–700 nm. Both Tur and BR smudges were characterized using a transmission electron microscope ((TEM) JEM-1400PLUS, JEOL USA, Inc., Peabody, MA, USA). Further characterization was done using Fourier-transform infrared spectroscopy (FTIR), Shimadzu FTIR-8300 spectrometer, San Diego, CA, USA) using potassium bromide (KBr) pellets. For FTIR, KBr was added directly into the beaker containing the oven-dried smudge collected from the Tur or BR fumes, and then the smudge was scraped along with the KBr powder. When the KBr powder took up the smudge, it turned grayish. After this, it was ground, and pellets were made as per routine FTIR analysis.

## 3. Results and Discussion

Explants obtained from wild growing ornamental plants were subjected to surface sterilization by fumigation. Three different methods were used, wherein method A, the growth medium was fumigated, in Method B, the explants themselves were fumigated, and in Method C, the explants were pretreated in a 20% ethanolic wash of the smudges from Tur or BR. The results from an evaluation of post inoculation contamination following fumigation and *in absentia* are shown in Figure 2. As observed from Figure 2, the graph presents the % contamination of the explants; it can be clearly seen that the results varied between the treatment methods and samples. However, as can be seen clearly from Figure 2, all three treatments were successful in leading to efficient surface sterilization of the explants compared to the untreated control. The observations recorded for each of the methods will now be discussed individually for the sake of clarity.

### 3.1. Surface Sterilization of Explants: Method A

In the case of turmeric-fumigated plates (TurFP), a nearly two-order magnitude decrease in surface contaminants was observed in the case of Explants 1 & 3; no fumigation effect was evident for Explant 2. With respect to BR-fumigated plates (BRFP), no fumigation effect was observed for all three explants compared to the untreated control. Method A was devised keeping in mind the antibacterial effect of Tur and BR smoke; however, merely fumigating the media and not the explants was insufficient to control the surface contaminants.

### 3.2. Surface Sterilization of Explants: Method B

Method B resulted in significant inhibition of microbial contaminants in all three explants in the case of Tur -fumigated explants (TurFE)) and BR-fumigated explants (BRFE). As observed from Figure 2, TurFE led to 65% inhibition and BRFE, 80% inhibition of surface contamination in Explant 1, in Explant 2, 80% (TurFE) and 20% (BRFE), and in Explant 3, absolute inhibition (100%) following TurFE fumigation and 80% inhibition with BRFE. These results indicate that Method B was indeed successful in the surface sterilization of the explants although varying degrees of sterilization were observed. These results indicate that there is scope for extension and improvement to arrive at absolute results.

### 3.3. Surface Sterilization of Explants: Method C

With Method C, Explant 1 showed only 16% contamination with Tur smoke suspension-treated explants (TurSSTE)) and 31% with BR smoke suspension-treated explants (BRSSTE); however, for explant 2, TurSSTE showed only 20% inhibition while BRSSTE succeeded with 50% inhibition. In Explant 3, 80% inhibition was obtained with TurSSTE pretreatment, and 100% inhibition of surface contaminants was observed with BRSSTE. With Method C, similar to method B, although we did observe variance, on the whole, it can be concluded that Method C was successful in leading to effective surface sterilization of the explants.

Summarizing the results, in the case of Explant 1, TurSSTE, followed by BRFE and BRSSTE were the most successful; explant 2: TurFE > BRSSTE > TurSSTE. In the case of Explant 3, both TurFE and BRSSTE led to absolute inhibition of surface contaminants, followed by BRFE and TurSSTE. Thus, it can be said that TurFE was the most effective surface decontamination method, followed by TurSSTE and BRSSTE. Figure 3A–C display the photographs of the culture plates, showing extensive fungal and bacterial contamination of the untreated controls (Figure 3A(a),B(a),C(a)), in the case of all three explants used in this study. Figure 3B,C portrays the significant inhibition of these contaminants as a result of fumigation. These experiments demonstrate the applicability of Tur (Figure 3A(b),B(b),C(b)) or BR (Figure 3A(c),B(c),C(c), respectively)-based fumigation for surface sterilization of explants in tissue culture.

### 3.4. Characterization of the Fumigants

The absorption properties of the Tur and BR smudges were characterized using UV–Vis spectroscopy. An absorption maximum was observed in the UV range of the spectrum between 220 and 250 nm for both the Tur and BR smudges (Figure 4A). This matches the absorption characteristics of carbon nanoparticles obtained from natural gas soot reported by previous researchers [29]. It was interesting to observe that these soot-derived carbon nanoparticles showed highly identical absorption characteristics. Further, FTIR was used to confirm the chemical nature and functionalization of the Tur/BR smudges (Figure 4B). Earlier workers [29,30,31] indicated that characteristic absorption bands of υ (O-H), υ (C-H), and υ (C=O) at 3320 cm^−1^, 2934 cm^−1,^ and 1764 cm^−1^, respectively, conform to the existence of –COOH in the case of carbon nanoparticles. The peak position at around 2360 cm^−1^ is due to the presence of CO_2_ and 1165 cm^−1^ the C=O (stretch). Thus, as observed from Figure 4B, the Tur and BR smudges showed absorption bands identical to those expected of carbon nanoparticles in smoke smudges, confirming the presence of nanoparticulate carbon in the smudges.

TEM was used to image the nanoparticulate carbon in the Tur and BR smudges; as observed from Figure 5A,B, many nanoparticulate carbon dots were evident. Carbon dots varying from 5 nm to 50 nm were well dispersed in the case of Tur (Figure 5A) and in BR, as seen from Figure 5B, particles smaller than 10 nm were observed. The larger particles were from fly ash debris. It is well established now that dispersion of carbon nanoparticles in ethanol lead to their uniform distribution [32]. Aggregation of nanoparticulate carbon is characteristic of carbon derived from smoke; the use of ethanol prevents this aggregation effect. The presence of carbon nanomaterial in smoke is not a new concept, but a well-established one. Reports of carbon nanomaterial present in soot are abundant and well documented; authors Anu and Manoj, 2012 [33], Song et al., 2011 [29], and Shooto et al., 2011 [34], have presented the characterization of such smoke-derived carbon nanomaterials [30,35]. These reports confirm the presence of carbon nanoparticles in smoke/soot. Our group has also reported the characterization and properties of soot-acquired carbon nanomaterials [30,31]. We have reported the use of fumigation for inhibition of biofilm on material surfaces [31]. Further, the role of carbon nanomaterials in turmeric smoke resulting in antimicrobial activity has also been reported [30]. Both these reports confirm the role of carbon in the fumigation effect.

### 3.5. Modulus Operandi of Tur and BR for Explant Surface Sterilization

Contemplating the reason why Tur/BR could lead to surface sterilization of the explants, we need to look into the composition of the fumigation sources. As for turmeric, it is a spice that is commonly used in Indian curries. It grows in India, China, and other Asian countries and has been used for centuries in traditional Indian medicine. Turmeric is well established for its powerful anti-inflammatory, antiseptic, antibacterial, antifungal, anticancer, and disinfectant properties and is used in several different ways, including burning and inhaling the smoke [36,37,38,39,40]. The burning of turmeric rhizomes and inhaling the smoke is a widespread practice in India. We have reported the effective inhibition of bacteria by the carbon dots functionalized with active turmeric ingredients in turmeric smoke [30,31].

The other fumigant source we used was benzoin resin (sambrani), which is a balsamic resin obtained from the bark of several species of trees in the genus *Styrax*. It is used in perfume incenses, as a flavoring, and for traditional medicine. Sambrani has been used in the culture for many years; it is a popular practice in Indian households to smoke this resin regularly as part of their customary and religious activity. Many countries use sambrani: in Arab countries, they use it directly, and in some countries they use it incorporated in incense. It is obtained by intentionally injuring the styrax tree. The plant is reported to produce this as a natural antimicrobial agent; benzoin begins as a semi-liquid sap oozing out from the injured bark. When dry, it becomes a somewhat pungent, dry (to slightly sticky) pale brown or rust-hued compound, having a somewhat marbled texture [41,42].

Thus, both the fumigation sources we have employed have a history of antimicrobial abilities. Moreover, fumigation as a source of cleansing or purging the air and atmosphere is a well-accepted fact [43]. Mohagheghzadeh et al. [44] summarized an exhaustive review of medicinal smoke, its sources, and uses, primarily highlighting the antimicrobial activity of smoke. Chun et al. [30] described and confirmed the nanoparticulate carbon role in the antimicrobial activity of smoke in their recent article. Our group [31] has also reported the successful inhibition of pathogenic biofilms using turmeric smoke-based fumigation. All these reports narrowed down the modulus operandi behind the antimicrobial disinfection of the carbon dots in the smoke. Based on these reports and the fact that our characterization studies revealed the presence of carbon material in the fumigant, we speculate that the mode of action of the surface sterilization of explants was because of the carbon nanodots functionalized with the active ingredients from the respective smoke, which led to the antimicrobicidal effect. Figure 6 represents this speculation. More studies are required to validate this speculation and to identify the exact mechanism behind the fumigation effect.

As observed from the results, it was found that fumigation of the explants and pretreatment of the explants in carbon soot solution resulted in excellent surface sterilization. Judy et al., (2016) recently reported the self-assembly of the soot-derived carbon dots onto substrates exposed to them [45]. It is a certain possibility that the carbon coated the explants when immersed in the soot solution and exposed to fumes. This carbon coating on the explants could have acted on the surface contaminants and sterilized these explants. Further, this carbon layer on the explants possibly acted as a protective coating on the explants, inhibiting any subsequent contaminant attack. Further, as reported in our previous work [30,31], the carbon nanodots in the soot were functionalized with the active ingredients present in the source material. Based on this, the antimicrobial activity expected of turmeric and BR was conferred onto the nanoparticulate carbon, which coated the explant surfaces, sterilizing the surface of microbial contaminants. Thus, the effect is not merely that from carbon nanoparticles, but apparently a combination of the antimicrobial properties of the active components (from Tur and BR) functionalized on them.

## 4. Conclusions

The successful use of turmeric and benzoin resin-based fumigation for the inhibition of surface contaminants on plant explants has been demonstrated for the first time. A surface-based interaction of the smoke-derived carbon nanomaterial with the explants is speculated as the reason for the surface decontamination of the tested explants. The carbon nanomaterial laden with the natural antimicrobial properties of the source material is expected to have brought about this surface sterilization effect. Surface sterilization as well as the fumigation of the culture environment is expected to have led to this outcome. This study takes the application a step further in that the use of toxic nanoparticles such as Ag, ZnO, and TiO_2_ routinely used for surface sterilization has been replaced by a more cytofriendly nanomaterial such as carbon.

## Figures and Tables

**Figure 1 ijerph-18-02282-f001:**
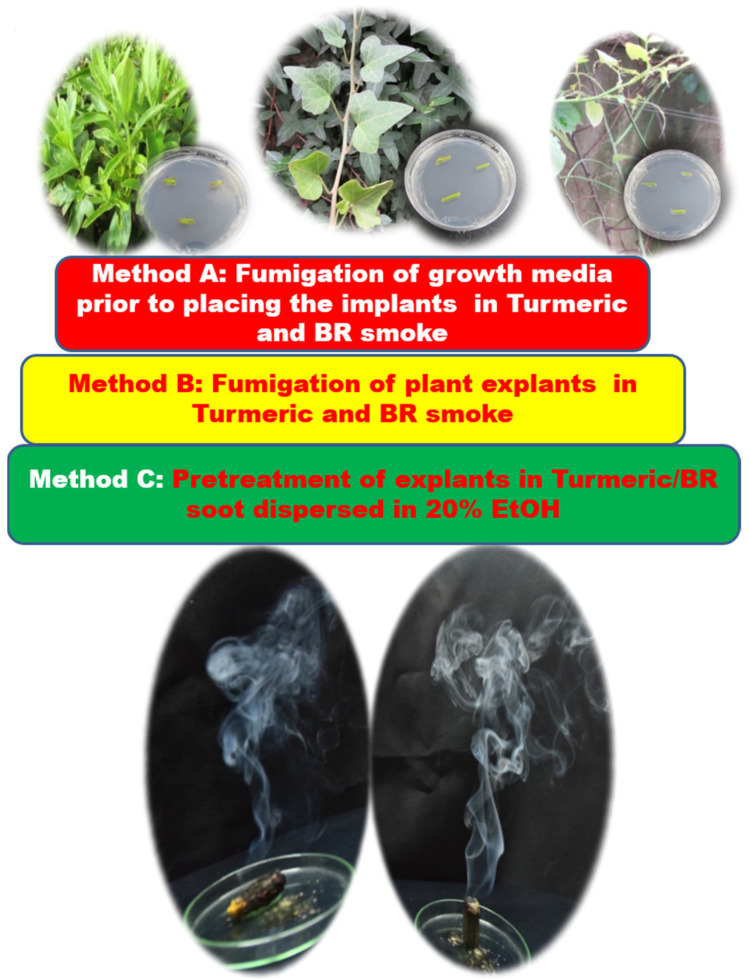
Schematic of the experimental methodology. BR—Benzoin resin

**Figure 2 ijerph-18-02282-f002:**
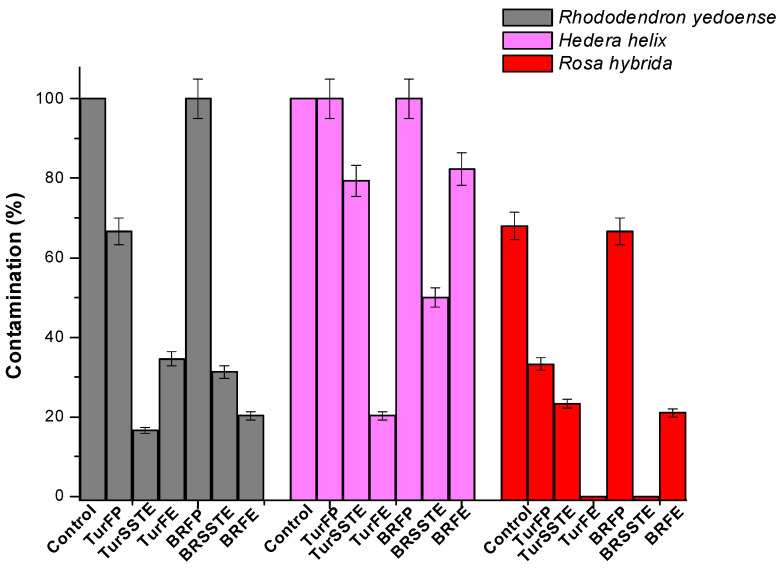
Graph showing results of the surface sterilization effect following fumigation.

**Figure 3 ijerph-18-02282-f003:**
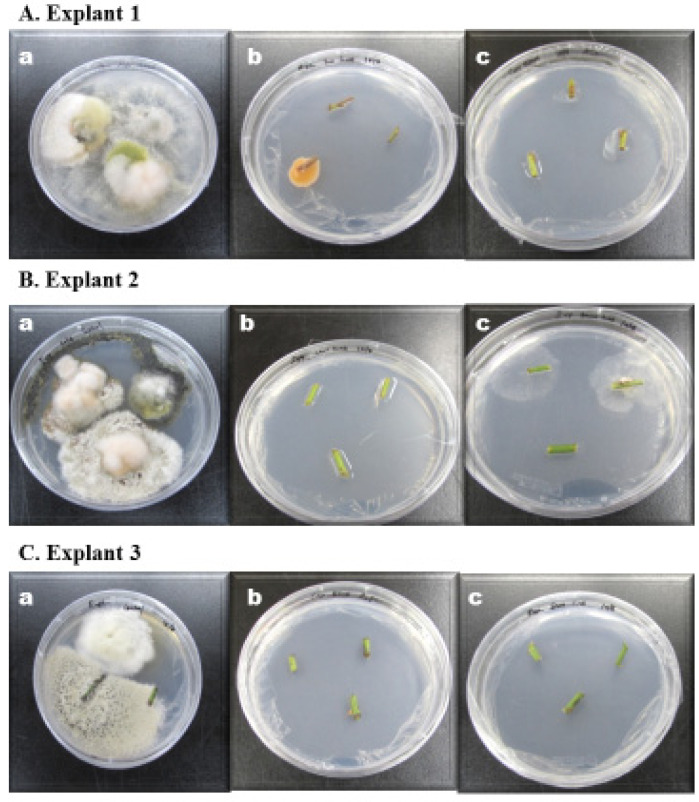
Photographs showing the results of the fumigation effect on explants versus the control. (**A**) Explant 1, (**B**) Explant 2, and (**C**) Explant 3; (a) Control (b) Tur (c) BR. Details are given in above text.

**Figure 4 ijerph-18-02282-f004:**
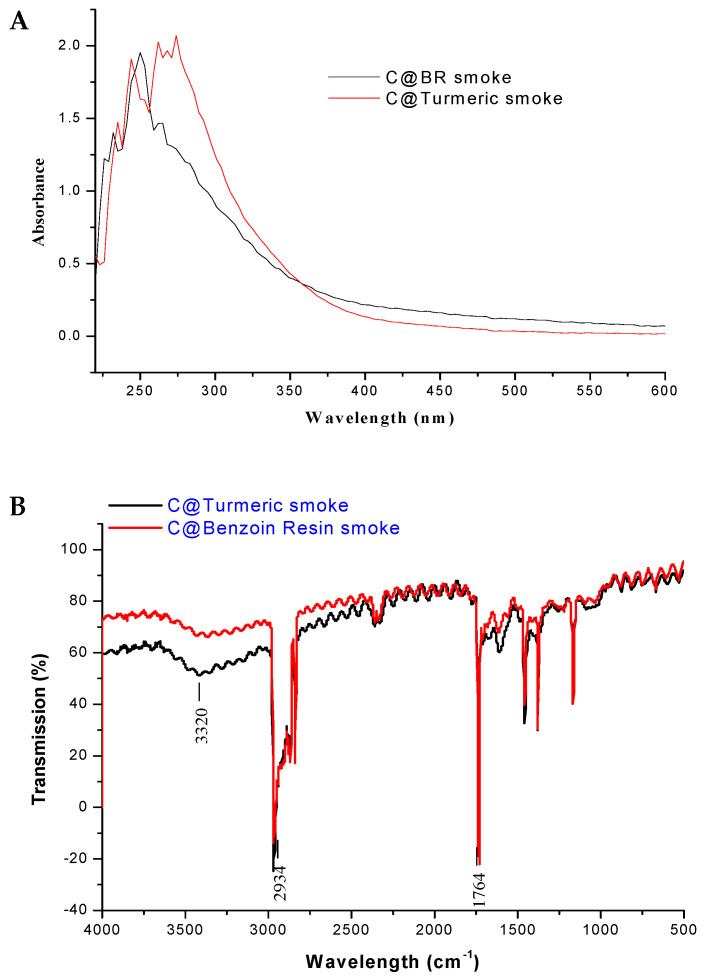
(**A**) UV–Vis curve of Tur and BR smoke. (**B**) FTIR of Tur and BR smoke.

**Figure 5 ijerph-18-02282-f005:**
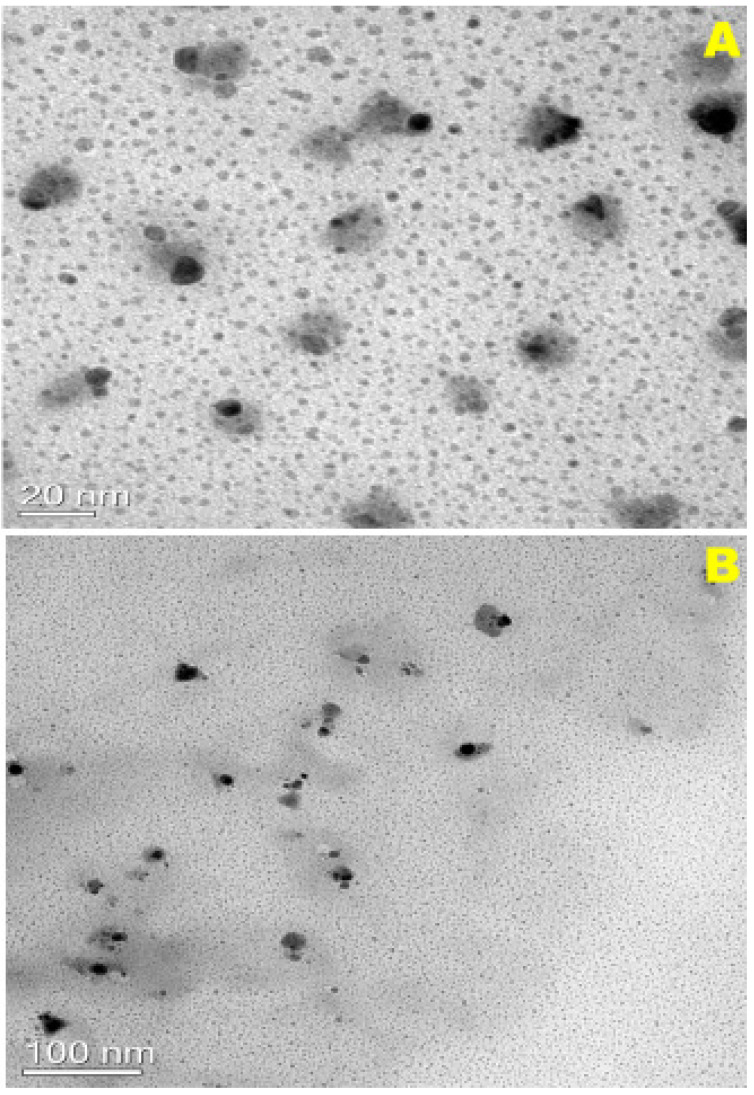
TEM micrographs (**A**) Turmeric and (**B**) BR (Benzoin resin) smudges.

**Figure 6 ijerph-18-02282-f006:**
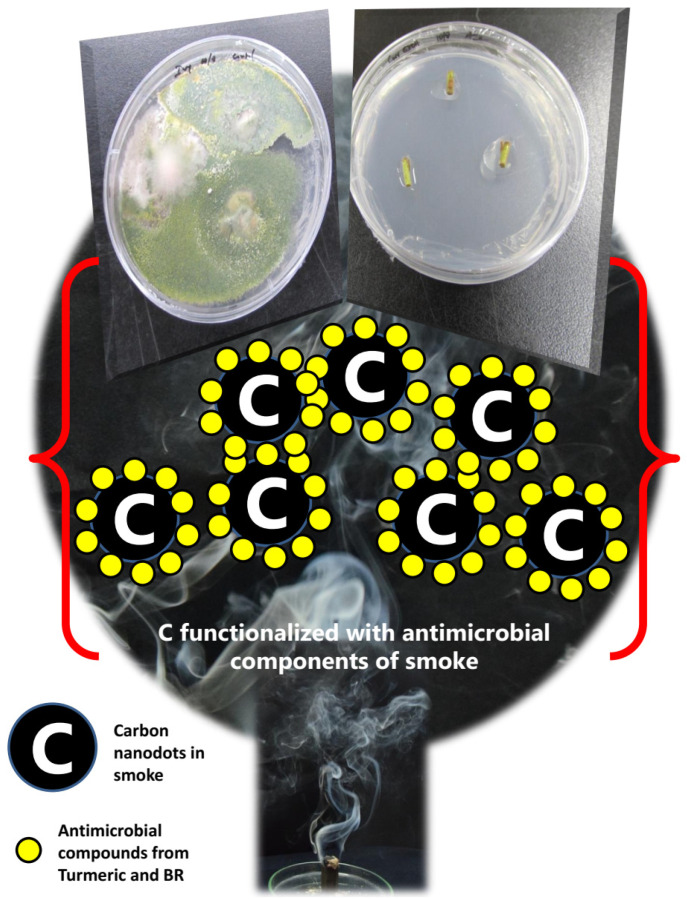
Cartoon speculating the mechanism of fumigation-based surface sterilization.

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
