# Peer review of "A Fumigation-Based Surface Sterilization Approach for Plant Tissue Culture"

_ijerph, 2021, doi:10.3390/ijerph18052282_

Round 1

Reviewer 1 Report

This a very interesting and promising manuscript concerning the sterilization by fumigation in tissue culture. Even the fact that there are a lot of speculations, the topic is important. On the other hand, the authors dident mention in what pathogens, this methods are adequate. They show the pathogens in the photo's but they dident specify them. Also, in results and discussion session, some parts should be rephrased and be more clear. Finally, the discussion of the results in some parts is limited. Thus, in my opinion, the manuscript should be reconsider for publication after major revision.

Below are my remarks:

Lines 31-70. This paragraph is not necessary and do not related with the purpose of the manuscript. In my opinion should be replaced by a small paragraph about plant tissue culture in general. 

Lines 71-81. Some relevant references should be added.

Line 90. "Some studies" please rephrase

Line 111. Give more details about mother plants i.e. age, pruning, plant protection, irrigation

Line 112. "campus" Please specified in details

Line 114 "rinsed with sterile distilled water". Specify the time for this

Line 122. "culture media plates" give more details

Line 126 "The 125 Tur/BR smoke were collected in sterile bottles". in what volume, concentration, etc. Also the bottles are from glass, plastic etc

Line 128 "Single nodal segments" ad i.e 1 cm long

Line 179-180 Please rephrased

Line 202. "Tur or BR based fumigation" (Fig. 3b,c, respectively). If this is correct, please ad

Line 205. Please specified A, B, C 

Line 226. "Figu" Please correct

Lines 239-265. These should be transferred to introduction section. It will be helpfull for the better understanding of the experiments. The rest of the paragraph should be rephrased

276 Judy et al.(19..)

Author Response

We humbly thank the editor and the reviewers for giving us an opportunity to revise the manuscript. Also we thank you for the valuable discussion and suggestions. We have now revised the manuscript as per your suggestions where ever applicable and have given a point by point response to all the comments raised. We have highlighted the changes in the manuscript using track changes. We thank you for the opportunity.

Reviewer 1

This a very interesting and promising manuscript concerning the sterilization by fumigation in tissue culture. Even the fact that there are a lot of speculations, the topic is important.

Ans. Thank you very much for the encouraging words of appreciation. We appreciate that and are really encouraged.

On the other hand, the authors dident mention in what pathogens, this methods are adequate. They show the pathogens in the photo's but they dident specify them.

Ans. In this case, we are merely demonstrating the surface sterilization ability of fumigations. The stems that we had taken were healthy plants, and the microbes on the surface are not necessarily pathogens. Hence, as a straightforward approach, we are merely showing the counts of surface contaminating microbes and the reduction in the surface contaminants following fumigations. Hope you understand that. Identification of the contaminants is beyond the scope of this study. Thank you for your kind understanding.

Also, in results and discussion session, some parts should be rephrased and be more clear. Finally, the discussion of the results in some parts is limited. Thus, in my opinion, the manuscript should be reconsider for publication after major revision.

Ans. Yes we do agree, that we need to discuss more elaborately, we have rephrased and revised these aspects in the revised text. Thank you.

Below are my remarks:

Lines 31-70. This paragraph is not necessary and do not related with the purpose of the manuscript. In my opinion should be replaced by a small paragraph about plant tissue culture in general. 

Ans. Yes agreed we have revised it. Thank you.

Lines 71-81. Some relevant references should be added.

Ans. References added.

Line 90. "Some studies" please rephrase

Ans. Rephrased.

Line 111. Give more details about mother plants i.e. age, pruning, plant protection, irrigation

Ans. Given details.

Line 112. "campus" Please specified in details

Ans. specified

Line 114 "rinsed with sterile distilled water". Specify the time for this

Ans. specified.

Line 122. "culture media plates" give more details

Ans. Details added.

Line 126 "The 125 Tur/BR smoke were collected in sterile bottles". in what volume, concentration, etc. Also the bottles are from glass, plastic etc

Ans. Details added

Line 128 "Single nodal segments" ad i.e 1 cm long

Ans. Specified.

Line 179-180 Please rephrased

Ans. Rephrased.

Line 202. "Tur or BR based fumigation" (Fig. 3b,c, respectively). If this is correct, please ad

Ans. Added.

Line 205. Please specified A, B, C 

Ans.specified

Line 226. "Figu" Please correct

Ans. corrected

Lines 239-265. These should be transferred to introduction section. It will be helpfull for the better understanding of the experiments. The rest of the paragraph should be rephrased

Ans. We feel it is easier to explain when retained in the experimental section, since it helps in the discussion. We have rephrased the paragraph. Thank you.

276 Judy et al.(19..)

Ans. Changed.

Reviewer 2 Report

As the author noticed that the plant tissue culture has led to breakthroughs in understanding and applying the fundamental knowledge towards harnessing more from plants. But the microbial contamination is a serious problems limiting the successful extrapolation of plant tissue culture practices. The common method for surface sterilization is done using chemicals, or toxic nano-materials. In this paper, the author try to use turmeric and benzoin resin based fumigation of explants. I think it is very clever way for fumigation. But the relationship between the experiment and final results is poor.

1) The turmeric and benzoin resin is "green"?

2) The only UV data and references supporting, I do not think it is enough evidences for nanoparticulate carbon analysis.

Within the current data, I cannot agree the final results. And also the topic is: Fumigation based ‘green’ surface sterilization approach for plant tissue culture: The nano carbon role in the fumigation ascendancy. But for me, it is hard to find enough discussion for this. 

Author Response

As the author noticed that the plant tissue culture has led to breakthroughs in understanding and applying the fundamental knowledge towards harnessing more from plants. But the microbial contamination is a serious problems limiting the successful extrapolation of plant tissue culture practices. The common method for surface sterilization is done using chemicals, or toxic nano-materials. In this paper, the author try to use turmeric and benzoin resin based fumigation of explants. I think it is very clever way for fumigation. But the relationship between the experiment and final results is poor.

Ans. We have increased the discussion aspect which will give a more clearer picture and validate the results. Thank you.

  • The turmeric and benzoin resin is "green"?

Ans. We understand your concern, but we have used this term because, these do not involve harmful chemicals, but plant based fumigation, Moreoever, these fumigants are also holding a reputation of possessing beneficial properties, so that’s why we deemed them green. But, either way since we felt u are not comfortable, we have removed the term green. Thank you.  

The only UV data and references supporting, I do not think it is enough evidences for nanoparticulate carbon analysis. Within the current data, I cannot agree the final results. And also the topic is: Fumigation based ‘green’ surface sterilization approach for plant tissue culture: The nano carbon role in the fumigation ascendancy. But for me, it is hard to find enough discussion for this. 

Ans. We do understand your concern, The fact that carbon is present in smoke is along established one, we have about three different investigations from our group that confirm the carbon aspect in smoke (Judy Gopal , Manikandan Muthu and Sechul Chun. Autochthonous self-assembly of nature's nanomaterials: green, parsimonious and antibacterial carbon nanofilms on glass Phys. Chem. Chem. Phys., 2016, 18, 18670-18677; S. Chun, M. Muthu, E. Gansukh, P. Thalappil and J. Gopal, Sci. Rep., 2016, 6, 35586. doi: 10.1038/srep35586a and        Anthonydhason, V., Gopal, J., Chun, S. et al. Nanocarbon Effect of Smoking Biofilms for Effective Control. J Clust Sci 29, 541–548 (2018). https://doi.org/10.1007/s10876-018-1394-2). We agree that these evidences have not been discussed adequately, we have now elaborated on these references in the discussion. Discussion has been added and results correlated with published proof. The revision will be a better projection. Thank you. Moreover we have modified the title removing the highlights on the nano carbon role. Thank you.

Reviewer 3 Report

this ms report some data about the effect of fumigation with deposit of carbon nano particles. the idea seems correct and the ms reasonably well prepared. Some problems should be improved, however, before going to publication.

first the presentation is made  in two parallel ways, printed text and cartoon. As is, the latter does not contain anything new and there is no indication of possible application.  In contrast   some data are perhaps worse than expected. As an example the IR spectra with such a strong beating of the armonics is suspect and the attribution weak. Finally, the applications suggested should be expanded, if, and only if, supported.

Author Response

We humbly thank the editor and the reviewers for giving us an opportunity to revise the manuscript. Also we thank you for the valuable discussion and suggestions. We have now revised the manuscript as per your suggestions where ever applicable and have given a point by point response to all the comments raised. We have highlighted the changes in the manuscript using track changes. We thank you for the opportunity.

this ms report some data about the effect of fumigation with deposit of carbon nano particles. the idea seems correct and the ms reasonably well prepared. Some problems should be improved, however, before going to publication.

Ans. Thank you for your encouraging words. We have addressed your concerns as much as possible. Thank you for your time and efforts.

first the presentation is made  in two parallel ways, printed text and cartoon. As is, the latter does not contain anything new and there is no indication of possible application.  In contrast   some data are perhaps worse than expected. As an example the IR spectra with such a strong beating of the armonics is suspect and the attribution weak. Finally, the applications suggested should be expanded, if, and only if, supported.

Ans. About the IR data, as is from all samples consisting of mixtures, we will expect noise, our previous manuscripts also show IR with such interferences, since the carbon from the smoke is not going to give smooth bands as would have isolated carbon nanoparticles, The smoke derived carbon have various functionalizations on it from the aromatic compounds from the smoke, essential oils too. These will surely show up as interferences in the IR bands. But the bands for carbon are clearly seen, we have now indicated the bands in the text and in the figure too.  Thus we are not drawing conclusion based on IR alone, we also have supporting TEM and UV and supplemented by other published references confirming the carbon in smoke. We have improved the revision, and added citations. Also confirming the presence of nanocarbon in smoke is already proven and that became our starting point for this study, we have discussed this in the discussion. Fig 6 is the only cartoon which speculates the mechanism, we have added a sentence that more fundamental investigations are needed to prove the mechanism behind the fumigation effect. This is something we are going ahead as future plan as an offshoot of this work, Thank you for your patience and understanding.

Round 2

Reviewer 1 Report

The manuscript was much improved and now is suitable for publication in the present form. Please see the remark below.

Fig. 3. Ad in the end of the legend (Details performed in the previous text paragraph) or something like that

Author Response

The manuscript was much improved and now is suitable for publication in the present form. Please see the remark below.

Fig. 3. Ad in the end of the legend (Details performed in the previous text paragraph) or something like that

And. we have added that details in the legend. thank you

Reviewer 2 Report

In this style, it is better than before, And more interesting for readers. So no further comments.

Author Response

In this style, it is better than before, And more interesting for readers. So no further comments.

Thank you very much for your encouraging remarks on our revision.